# Lectins in Cervical Screening

**DOI:** 10.3390/cancers12071928

**Published:** 2020-07-16

**Authors:** Anita WW Lim, André A. Neves, Sarah Lam Shang Leen, Pierre Lao-Sirieix, Elizabeth Bird-Lieberman, Naveena Singh, Michael Sheaff, Tony Hollingworth, Kevin Brindle, Peter Sasieni

**Affiliations:** 1Wolfson Institute of Preventive Medicine, Centre for Cancer Prevention, Barts & The London School of Medicine and Dentistry, Queen Mary University of London, Charterhouse Square, London EC1M 6BQ, UK; peter.sasieni@kcl.ac.uk; 2School of Cancer and Pharmaceutical Sciences, Faculty of Life Sciences and Medicine, King’s College London, London SE1 9RT, UK; 3Cancer Research UK Cambridge Institute, Li-Ka Shing Centre, University of Cambridge, Cambridge CB2 0RE, UK; andre.neves@cruk.cam.ac.uk (A.A.N.); kmb1001@cam.ac.uk (K.B.); 4Department of Cellular Pathology, Barts and the London NHS Trust, Pathology and Pharmacy Building, The Royal London Hospital, 80 Newark Street, London E1 2ES, UK; sarah.lamshang@nhs.net (S.L.S.L.); singh.naveena@nhs.net (N.S.); michael.sheaff@nhs.net (M.S.); 5MRC Cancer Cell Unit, Hutchison-MRC Research Centre, Cambridge CB2 0XZ, UK; pierre.lao-sirieix@astrazeneca.com (P.L.-S.); Elizabeth.Bird-lieberman@ouh.nhs.uk (E.B.-L.); 6Translational Gastroenterology Unit and Biomedical Research Centre, John Radcliffe Hospital, Headley Way, Headington, Oxford OX3 9DU, UK; 7Whipps Cross University Hospital, Barts Health NHS Trust, Whipps Cross Road, London E11 1NR, UK; t.hollingworth@qmul.ac.uk

**Keywords:** lectins, cervical cancer, screening, glycans

## Abstract

Cervical screening in low-resource settings remains an unmet need. Lectins are naturally occurring sugar-binding glycoproteins whose binding patterns change as cancer develops. Lectins discriminate between dysplasia and normal tissue in several precancerous conditions. We explored whether lectins could be developed for cervical screening via visual inspection. Discovery work comprised lectin histochemistry using a panel of candidate lectins on fixed-human cervix tissue (high-grade cervical intraepithelial neoplasia (CIN3, *n* = 20) or normal (*n* = 20)), followed by validation in a separate cohort (30 normal, 25 CIN1, 25 CIN3). Lectin binding was assessed visually according to staining intensity. To validate findings macroscopically, near-infra red fluorescence imaging was conducted on freshly-resected cervix (1 normal, 7 CIN3), incubated with topically applied fluorescently-labelled lectin. Fluorescence signal was compared for biopsies and whole specimens according to regions of interest, identified by the overlay of histopathology grids. Lectin histochemistry identified two lectins—wheat germ agglutinin (WGA) and *Helix pomatia* agglutinin (HPA)—with significantly decreased binding to CIN3 versus normal in both discovery and validation cohorts. Findings at the macroscopic level confirmed weaker WGA binding (lower signal intensity) in CIN3 vs. normal for biopsies (*p* = 0.0308) and within whole specimens (*p* = 0.0312). Our findings confirm proof-of-principle and indicate that WGA could potentially be developed further as a probe for high-grade cervical disease.

## 1. Introduction

Cervical cancer remains the most common female malignancy in many developing countries [1] where high-quality organised screening has not been possible [2]. Of the ~570,000 women diagnosed each year globally, over half will die from the disease—88% of these deaths occur in developing countries [1]. Cervical screening involves detecting and treating high-grade cervical intraepithelial neoplasia (CIN grade 2 or worse, CIN2+) to prevent cancer from developing. Due to financial constraints and limited infrastructure, many low resource countries favour cervical screening via visual inspection (with acetic acid (VIA) and/or Lugol’s iodine (VILI), topically applied to the cervix making high-grade disease visible) [3]. Visual inspection enables women to be seen and treated at the same visit but has much room for improvement due to its low specificity (14–98%) and variable sensitivity (41–98%) to detect CIN2+ [4,5]. Given the heavy and disproportionate burden of disease in developing countries, there is a need to identify a specific, reproducible and affordable imaging method that can be used for cervical screening in low-resource settings. In particular, there is a need for visualization methods to identify precursor lesions on the cervix.

Lectins are naturally occurring glycoproteins of non-immune origin which bind to sugars (glycans) on the surface of cells [6]. Their ubiquity throughout nature means that they are widely available and can be produced at low cost. The neoplastic transformation of cells is associated with changes in cell surface glycoconjugates [7,8] and alterations in lectin binding with dysplasia or carcinogenesis have been observed in numerous organs (e.g., oesophagus [9], pancreas [10], skin [11,12], bladder [13,14], colon [15], stomach [16,17,18] and cervix [19]). Aberrant sialylation and fucosylation have also been described as early markers of cervical carcinogenesis [20]. There is compelling evidence that topically applied fluorescently labelled lectins can identify areas of dysplasia. WGA binding decreases with increasing dysplasia in Barrett’s oesophagus (a premalignant state of oesophageal adenocarcinoma) [9,21], colorectal dysplasia [22] and oral dysplasia and malignancy [23,24,25,26]. In cervical cancer, markedly reduced sialylation was found in the glycoproteins of malignant tissue [20].

The aim of this study was to act as a proof-of-principle study to explore whether fluorescently-labelled lectins, topically applied to the cervix, show potential to be developed as a form of cervical screening in low-resource settings. This lectins-based approach to cervical screening would comprise real-time visual assessment using wide-field fluorescence imaging, fluorescence-enhanced colposcopy or wide-field fluorescence imaging, based on a fluorescence-capable light source (e.g., torch) [27,28]. Wheat germ agglutinin (WGA) demonstrated an ability to discriminate normal tissue from high-grade CIN at both the microscopic (lectin histochemistry on formalin-fixed paraffin-embedded (FFPE) tissue) and, to a limited extent, the macroscopic level (near-infrared (NIR) fluorescence imaging of conjugated lectins topically applied to freshly resected human cervix). This exploratory work warrants further in vivo studies with WGA to further explore its potential in cervical screening.

## 2. Results

### 2.1. Discovery Lectin Histochemistry

Eleven candidate lectins were identified by a literature search and by the previous work in Barrett’s oesophagus [9]. Of these, eight were successfully optimised and taken forward for lectin histochemistry. Lectin staining was significantly less intense in 20 CIN3 slides compared with 20 slides of normal squamous epithelium tissue for three of the eight candidate lectins: Wheat germ agglutinin (WGA) (*p* < 0.0001); *Helix pomatia* agglutinin (HPA) (*p* = 0.0002); *Urea europaeus* agglutinin (UEA) (*p* = 0.007). Representative examples of lectin staining for these and corresponding lectin staining scores are shown in Figure 1a,b.

For UEA, the absolute staining intensity was weak (median score of 1 in normal cervical tissue, where staining intensity was scored on an arbitrary scale: 0 = no stain; 1 = weak stain; 2 = moderate stain; 3 = strong stain).

### 2.2. Validation of Lectin Histochemistry

Validation was undertaken for WGA and HPA (the best discriminators of dysplasia during the discovery phase) in a separate cohort of FFPE samples (30 normal, 25 CIN1, 25 CIN3). UEA was not taken forward for validation due to the weak absolute staining intensity in the discovery phase, indicating it would not be useful as a visual assessment tool. As with the discovery cohort, lectin staining was weaker in CIN3 versus normal tissue for both WGA (*p* = 0.0044) and HPA (*p* = 0.0002). However, the differences in staining intensity were less marked than in the discovery cohort—median scores of 0 (CIN3) and 2.75 (normal) versus 1 (CIN3) and 2 (normal) for WGA and HPA, respectively.

The staining of CIN1 was similar to normal tissue but more intense than the staining of CIN3 with HPA (*p* = 0.0008), but not with WGA (*p* = 0.6943). Lectin staining scores for WGA and HPA for normal cervix, CIN1 and CIN3, are shown in Figure 2.

Within specimen comparisons of lectin, staining was also performed for normal tissue adjacent to CIN1 and CIN3. The staining of normal tissue adjacent to CIN was consistently stronger than CIN3 (WGA *p* = 0.0487, HPA *p* = 0.0001) and CIN1 (WGA *p* = 0.0023 and HPA *p* = 0.0028).

Overall concordance for inter-rater agreement for lectin staining for discovery and validation was 86.7% (kappa 0.65 (95%CI 0.60–0.67), *p* < 0.0001).

#### Lectin Histochemistry Sensitivity and Specificity for CIN3

In order to further assess the discriminatory power of lectin histochemistry to discern CIN3 from normal cervix, we calculated sensitivity and specificity using a staining score threshold of 0/1 = “positive” and 2/3 = “negative”. In the discovery cohort, WGA demonstrated the highest discriminatory power with a sensitivity of 75% (15/20) (95%CI 50.9–91.3), a specificity of 85% (17/20) (95%CI 62.1–96.8) and an Area Under the Curve (AUC) of 0.884 (95%CI 78.2–98.5). For HPA, sensitivity was 70% (14/20) (95%CI 45.7–88.1), specificity was 80% (16/20) (95%CI 56.3–94.2) and AUC was 0.843 (95%CI 0.710–0.976), and for UEA, sensitivity was 85% (17/20) (95%CI 62.1–96.8), specificity was 40% (8/20) (95%CI 19.1–63.9) and AUC was 0.796 (95%CI 0.660–0.932).

In the validation cohort, the sensitivity and specificity were lower in comparison to the discovery cohort for both lectins. WGA sensitivity was 60% (15/25) (95%CI 38.7–78.9), specificity was 87% (26/30) (95%CI 69.3- 96.2), and AUC was 0.708 (95%CI 0.566–0.850), while HPA sensitivity was 84% (21/25) (95%CI 63.9–95.5), specificity was 69% (20/29) (95%CI 49.2–84.7) and AUC was 0.792 (95%CI 0.661–0.922). Figure 3 shows the corresponding Receiver–Operator Characteristic (ROC) curves for discovery and validation cohorts.

### 2.3. Molecular Imaging of Topically Applied Fluorescently-Labelled Lectin to Freshly Resected Human Cervix

Only WGA was taken forward for ex vivo studies of molecular imaging. The decision to not take HPA forward was mainly based on the favourable existing data on WGA in Barrett’s oesophagus [9] and colorectal cancer [21]. Additionally, HPA is derived from the Burgundy snail, which is known to be an allergen [29], limiting its potential as a topical agent. Other lectins have been used as histochemical reagents for sialic detection (e.g., GS-II, TJA, SNA, MAL) [30]. However, their reduced availability and potential toxicity make them less viable as candidates for topical imaging applications in comparison with WGA, which is derived from edible wheat germ, a dietary constituent. Freshly resected cervix tissue samples were obtained from nine women. Two samples were pinned incorrectly (i.e., ectocervix facing down), leaving a total of seven samples that were analysable (1 normal and 6 CIN3). From these, 22 biopsies were taken, of which 14 were analysable—normal (*n* = 10) and CIN3 (*n* = 4). Excluded biopsies comprised samples from two women (4 biopsies from each). One had a large transformation zone and none of the biopsies contained tissue relevant for analysis (i.e., no normal squamous epithelium or CIN3, endocervical/metaplastic epithelium only) and, in the other sample, the biopsies were embedded incorrectly, hence histopathology could not be read.

#### 2.3.1. Whole Specimens

Pairwise (within sample) analysis of regions of interest (ROI) within six whole specimens, as shown in Table 1 (patients 104–110), found a significant difference in the mean MFI (*p* = 0.0312) between regions of CIN3 and regions of normal tissue.

#### 2.3.2. Biopsies

The comparison of signal intensity for all 14 biopsies from the whole specimens, according to the histology (of the biopsy), showed that WGA binding was significantly lower in CIN3 versus normal biopsies; mean MFI 0.066 (±0.011) vs. 0.104 (±0.033), respectively (t (13.89) = 3.37, *p*= 0.0046).

The final two samples collected were specifically targeted as they had large and demarcated areas of CIN3 (on colposcopic impression) with large areas of surrounding normal tissue. These samples were analysed whole, with no biopsies collected. One of the specimens had a particularly large area of CIN3 with normal surrounding squamous epithelial tissue, as shown in Figure 4. In this specimen, WGA binding appeared to be weaker in the CIN3 area, as demonstrated by lower fluorescence signal. Figure 4e shows that the large area of CIN3 appeared substantially darker (less fluorescence) in the ROI compared with the surrounding normal tissue.

## 3. Discussion

This exploratory study has demonstrated proof-of-principle that topically applied fluorescently-labelled WGA lectin could potentially be developed as an imaging probe for real-time cervical screening using wide field fluorescence imaging. Lectin histochemistry in the discovery cohort was particularly promising and showed the remarkable ability of lectins WGA and HPA to discriminate CIN3 from normal squamous epithelium. The findings were confirmed in the validation cohort but were less distinct for reasons that are unclear. Validation at the macroscopic level ex vivo were also positive. In a pooled analysis of biopsies fluorescently labelled, WGA binding was significantly lower (lower signal intensity) for CIN3 versus normal tissue ex vivo. Similarly, signal intensity for regions of interest within whole specimens showed significantly lower signal from CIN3 compared to normal tissue (*n* = 6, *p* = 0.0312). On lectin histochemistry CIN1 staining was less intense than CIN3 for HPA but was similar for WGA.

It is encouraging that WGA was identified as having the highest discriminatory power from the panel of lectins, given that this is the same lectin that has been shown to identify dysplasia in other precancerous conditions involving epithelial tissue [9,21,22]. WGA was able to consistently delineate areas of high grade disease within the same specimen, as shown in Table 1, albeit with low contrast. The latter ability was reassuring as this most closely replicates the clinical scenario in which lectins would be used for cervical cancer screening (i.e., topically applied lectin followed by biopsy of suspected CIN3 based on visualization of signal intensity). The extent of contrast obtained might have been affected by field effects and specimen quality. Most of the CIN3 samples collected had only very small areas of CIN3, which tended to be within the crypts and therefore did not lend themselves well to visual assessment. Surrounding areas of tissue were also often suboptimal with a large transformation zone and/or cervical ectopy leaving little to no normal squamous epithelium for comparison. Nevertheless, results from the validation cohort were promising and validated observations from the discovery cohort.

Key limitations of our study are the small number of samples included and the intrinsically imprecise co-registration of histopathology with fluorescent images acquired from fresh specimens. Although the co-registration of histopathology grids with fluorescent images by image overlay is prone to error, this is, nevertheless, a suitable approach for exploratory studies [21]. Similarly, the scoring for the lectin histology staining in the discovery cohort was based on subjective visual assessment. As mentioned above, the qualities of most of the ex vivo specimens collected were suboptimal for our chosen method of visual assessment and may explain the results. In well-screened populations (such as England), cervical disease with large areas of CIN3 are rare, thus suitable samples were difficult to obtain.

In previous studies of lectin histochemistry in FFPE human cervical tissue, *Griffonia simplicifolia* II was the only lectin from a panel of nine which showed significantly different binding patterns between normal and CIN, but no difference was found using WGA [31,32]. These studies, however, were performed with a majority of specimens containing CIN 1–2. Our findings are in accord with other studies using fluorescently-labelled WGA in the gastrointestinal tract, which demonstrated the lower binding of WGA with increasing grades of dysplasia [9,22]. Similarly, in Barrett’s oesophagus, WGA showed decreased binding in dysplastic versus non-dysplastic areas in ex vivo studies [9,21]. WGA-IR800 was able to identify dysplasia at the whole specimen level in freshly collected endoscopic mucosal resections in both pooled analyses across specimens and within specimen analyses [21].

Aberrant glycosylation has been explored as a biomarker for carcinogenesis in several cancers [7]. One of the most advanced lectin probes in terms of clinical translation is a chairside optimal imaging system using WGA-fluorescein isothiocyanate (FITC) with a handheld light to detect oral malignancies [23,24]. Baeten et al. [23] found that topically applied WGA-FITC to oral mucosa in vivo had a sensitivity of 89% (95% CI 73–96%) and specificity of 82% (95% CI 62–93%) for detecting oral dysplasia and cancer. This model in oral cancer is highly relevant to the future development of lectins as a probe for cervical screening, given that both malignancies arise in squamous epithelium and are particularly prevalent in developing countries in sub-Saharan Africa and South East Asia [33,34]. Furthermore, a handheld light device is likely to be easily translated into cervical screening for low-resource settings where lightweight mobile equipment is desirable [35].

Currently, the majority of tools being developed for cervical screening in low resource settings remain focused on improving “see-and-treat” approaches (i.e., real time visualization of disease) [27,36,37]. Such approaches can be used either for primary screening or for triage of HPV positive women. A key advantage of “see-and-treat” cervical screening modalities is that women can be screened and treated (i.e., CIN2+ removed by excision or cautery) at the same visit. Digital cervicography [38] and other forms of digital imaging, such as smartphone-based images (often combined with machine-learning) [36,39,40], are being developed as either improvements or adjuncts to “see-and-treat” approaches. The emerging field of smartphone fluorescence microscopy [28] aligns well with these developments and with our proposed approach. Our early exploratory data warrant further studies of WGA to see if it can be successfully developed as an imaging biomarker for “see-and-treat” early cervical cancer screening. WGA has binding specificity for terminally installed sialic acid residues on tissue glycoproteins. [20,41]. Moreover, cervical cancer has lower levels of sialylation in comparison with healthy cervix [41], and, therefore, the potential for a fluorescently-labelled version of the lectin to be used topically as an early diagnostic imaging probe is worthy of further investigation. Even if the discriminatory ability shown in the small number cohort analysed is lower than hoped, potentially, a lectin probe could be used in conjunction with VIA and/or VILI (and have a role in helping to define whether high grade disease is present (after first defining an area of interest using acetic acid/Lugol’s iodine).

A further potential application of lectins in cervical screening is in colposcopy. The main goals of colposcopy are to identify areas most likely to be high grade CIN (for diagnostic biopsy) and to identify areas of CIN (in women with confirmed high-grade CIN) for excisional or destructive treatment. Acetic acid and Lugol’s iodine are used in situ to identify cervical neoplasia for diagnostic biopsy. The accuracy of colposcopically-guided biopsy has been criticised for its low sensitivity for detecting CIN grade 2 or worse [42,43,44]. An imaging probe with higher sensitivity and adequate specificity for detecting high grade CIN than acetic acid and Lugol’s iodine remains an unmet clinical need.

## 4. Materials and Methods

### 4.1. Human Tissue

Lectin histochemistry was performed on 120 formalin fixed paraffin embedded (FFPE) cervical tissue samples. Ethical approval was obtained from the South West Wales REC (13/WA/0151). All samples were provided anonymously from a tissue bank of diagnostic specimens at the Barts Health NHS Trust, London, UK. Ten sections (5 μm thick) were cut from each tissue block and mounted onto slides for staining. The fifth section from each set was stained with haematoxylin and eosin (H&E). An independent histopathologist (S.L.S.L.) confirmed the clinical diagnosis and marked out areas of dysplasia on the H&E sections of CIN1 and CIN3 samples. Forty samples were used for the discovery work—20 normal and 20 CIN3 samples. Eighty samples were included for the validation work—30 normal, 25 CIN1 and 25 CIN3. Samples were derived from tissue that had been removed in the previous 5 years, from women aged between 25 and 59 years at the time of resection.

For the molecular imaging of lectin staining, ethical approval was granted from the Yorkshire and the Humber–Bradford Leeds Research Ethics Committee (15/YH/0371). Freshly resected cervix tissue was prospectively collected from nine women for treatment purposes from a single secondary care centre between January 2016 and June 2017. Written informed consent was obtained from all participants. Normal cervical tissue comprised surplus surgical tissue from hysterectomies performed for non-cervical indications (e.g., fibroids, endometriosis). During the study, two “normal” cervix tissue samples were collected; however, an unexpected finding of a small area of CIN3 was identified during histopathology reading on one of the samples. This sample was reclassified into the CIN3 group for analysis. Seven high-grade CIN tissue samples were collected from women undergoing excisional treatment (knife cone or loop biopsies) for confirmed CIN3 on previous biopsy. Participants had a mean age of 39.7 years (range, 25–73 years), (normal age range 53–73, CIN3 age range 25–44 years). The first seven samples collected had only small areas of CIN3 (often within the crypts and not on the surface) and very little surrounding normal squamous epithelial tissue for comparison (due to large transformation zones and/or ectropions). Therefore, for the final two samples, we specifically targeted women with large areas of CIN3 on colposcopy, and without an ectropion or large transformation zone.

### 4.2. Pathological Diagnosis

Histopathological assessment for all specimens was carried out (independently of fluorescence staining) at Barts Health NHS Trust (Royal London Hospital, London). For each cervix sample, the specimen was photographed and dissected by a histopathologist (S.L.S.L. or N.S.), and then correlated with the H&E stained sections to map the areas of normal squamous epithelium and areas of CIN3 onto the photograph to produce a “histopathology grid”. For lectin histochemistry, N.S. and S.L.S.L. examined the H&E stained sections to determine histology. A single independent histopathologist (either S.L.S.L. or N.S.) examined the slides for the fluorescently-labelled lectin samples.

### 4.3. Identification of Candidate Lectins

A panel of 11 candidate lectins were identified for initial screening. Eight [31,32,45,46,47,48,49,50,51,52,53,54,55,56,57,58,59] were identified via a literature search: *Concanavalin ensiformis* (ConA) [31,49,52,54,60]; *Griffonia simplicifolia II* (GS II) [31,32,59]; *Vicia villosa* agglutinin (VVA) [51]; *Artocarpus integrifolia*-Jack fruit lectin (JFL) [55,57,58]; *Sambucus nigra* (SNA) [53]; *Maackia amurensis* (MAA) [53]; *Ulex europaeus* I (UEA I) [31,32,45,47,48]; *Dolichos Biflorus* agglutinin (DBA) [52,59]. The other three lectins were those which best identified dysplasia in Barrett’s oesophagus [9]: *Triticum vulgaris* (wheat germ agglutinin (WGA)); *Helix pomatia agglutinin* (HPA); *Aspergillus oryzae* lectin (AOL) [9].

### 4.4. Lectin Histochemistry

For the discovery work, lectin histochemistry was performed using eight candidate lectins (DBA, Con A and MAA could not be optimised (no real positivity)) on FFPE samples (20 normal and 20 CIN3). Validation was undertaken in a separate cohort of biopsies (30 normal, 25 CIN1 and 25 CIN3) for WGA and HPA.

Slides were de-paraffinized by immersion in xylene (2 × 2 min), followed by immersion in decreasing concentrations of ethanol (96%, 75%, and 50%) (× 5 min), and washed in water (2 × 5 min) then lectin buffer (20 mM HEPES, 150 mM NaCl, 1 mM MgCl2, 1 mM CaCl2, pH 7.4) (1 × 5 min). Slides were then placed in a humidified incubation chamber and 5 μg/mL lectin (labelled with biotin) was applied, followed by incubation at 37% for 15 min. Unbound lectin was washed off by immersion in running water (2 × 30 min), and mounted with Prolong Gold antifade reagent with DAPI (Invitrogen, Paisley, UK). Biotin labelled lectin was visualised with the standard Avidin–Biotin Complex (ABC) method with a complex of avidin–biotin peroxidase. The peroxidase was then developed by 3,3’ diaminobenzidine (DAB) to produce different colorimetric end products.

#### 4.4.1. Scoring

For the discovery work, two researchers (including at least one histopathologist, N.S., S.L.S.L. and P.L.S.) independently scored the top third of squamous epithelium for the intensity of lectin staining. A histopathologist (M.S.) adjudicated when there was disagreement. Each section was scored for staining intensity on an arbitrary scale: 0 = no stain; 1 = weak stain; 2 = moderate stain; 3 = strong stain. For sections from normal cervix a score for the entire section was taken. For sections with CIN3, an average of scores across all areas of CIN3 was calculated. Additionally, areas of normal tissue on CIN3 sections were scored within section comparisons of normal versus CIN3 lectin staining. 

Concordance for inter-rater agreement was calculated for lectin staining scores with the associated weighted kappa statistic.

#### 4.4.2. Sensitivity and Specificity to High Grade Cervical Disease

The sensitivity and specificity of CIN3 for lectin histochemistry staining was calculated for lectins, which showed discriminatory power using a threshold score of 0/1 = positive and ≥2 = negative.

### 4.5. Molecular Imaging of Fluorescently-Labelled Lectin

#### 4.5.1. Staining Protocol

Samples were transported at 2–8 °C (on wet ice and/or cool packs) to a clinic in Cambridge University Hospitals NHS Foundation Trust, where each cervical specimen was pinned under an open-field fluorescent imaging camera (Fluobeam^®^ 800 CE Mark, Fluoptics, Grenoble, France), sprayed with fluorescein-labelled WGA (WGA-IR800CW, was prepared as described previously [21]) and incubated by immersion in lectin solution (10 ug/mL in PBS buffer) for up to 15 min at room temperature. The tissue was then washed three times with excess cold PBS buffer (to remove any excess unbound lectin) and imaged using near-infrared fluorescent (NIR) and visible light.

Using NIR fluorescence as a guide, up to four biopsies (4 mm diameter) were collected from each specimen (one from each quadrant, using a disposable skin biopsy punch device). From CIN3 samples, two were collected from regions of lowest fluorescent signal intensity (i.e., suspected dysplasia) and two from the highest intensity regions (i.e., expected to be normal). The researchers collecting the biopsies were guided purely by visual impression of signal intensity and had no prior knowledge of where the areas of CIN3 were located. For normal samples, four punch biopsies were taken from random areas in the transformation zone of the cervix (one from each quadrant). The punch biopsies and the original specimens were then fixed in formalin and sent for routine histological assessment. The mean time from excision to fixation was 2 h 35 min for the first protocol and 33 min for the second laboratory protocol.

The laboratory protocol was changed after seven samples were analysed, so that lectin incubation was immediately performed after resection, washed to remove excess lectin then and fixed in formalin for 24 h and immersed in ethanol. Freshly excised tissue samples were pinned onto a specimen card and incubated with fluorescein-labelled WGA lectin for up to 15 min at room temperature and then washed with cold buffer (to remove any excess unbound lectin). Samples were then fixed in formalin for 24 h and then immersed in 70% ethanol. Specimens were imaged and analysed whole (i.e., no biopsies were taken).

Each specimen was pinned under a Fluobeam^®^ 800 camera and imaged using near-infrared fluorescent and visible light. Samples were then sent for routine histological assessment. In addition, at the time of sectioning for the reading of histopathology (prior to H&E staining), one extra section was taken at each level, slide-mounted and left unstained. These slides were fluorescence imaged and compared (overlaid) to the corresponding H&E stained slides.

#### 4.5.2. Quantification of NIR Fluorescence Signal for Imaging Data

The NIR fluorescence signal was quantified for regions of interest (ROI) as the mean lectin fluorescence intensity (MFI), following the subtraction of the background fluorescence. An average MFI was calculated to produce a single score for analysis for: (i) areas of CIN3 as demarcated by the histopathology grid overlay; (ii) each biopsy according to histopathology; (iii) whole specimens (according to overall specimen histology). MFI was standardised for exposure time, light intensity and distance to imaging.

### 4.6. Analysis

The Wilcoxon rank sum test was used to compare lectin staining in normal epithelium versus CIN3, which was determined by comparison with an adjacent H&E stained section.

Concordance for inter-rater agreement for lectin staining was assessed using the weighted Cohen’s kappa.

The Student’s t test with Welch’s correction for unequal variances was used to compare mean fluorescence for comparing regions of interest within whole specimen ex vivo samples. Pairwise comparison of CIN3 versus normal cervix biopsies was performed using the two-tailed Wilcoxon matched-pairs signed rank test.

All statistical analyses were performed using Statistical Software: Release 16. StataCorp, 2019 (College Station, TX, USA: StataCorp LP). A *p* value of less than 0.05 was considered statistically significant. All statistical tests were two-sided.

## 5. Conclusions

As imaging probes, lectins are particularly appealing because their binding specificities are well characterized, they are widely available, relatively cheap and often have low toxicities (if applied topically). To our knowledge, this is the first evidence that topically applied lectins show the potential to be developed as a visual assessment tool for cervical screening. Our findings warrant further evaluation in an in vivo setting.

## Figures and Tables

**Figure 1 cancers-12-01928-f001:**
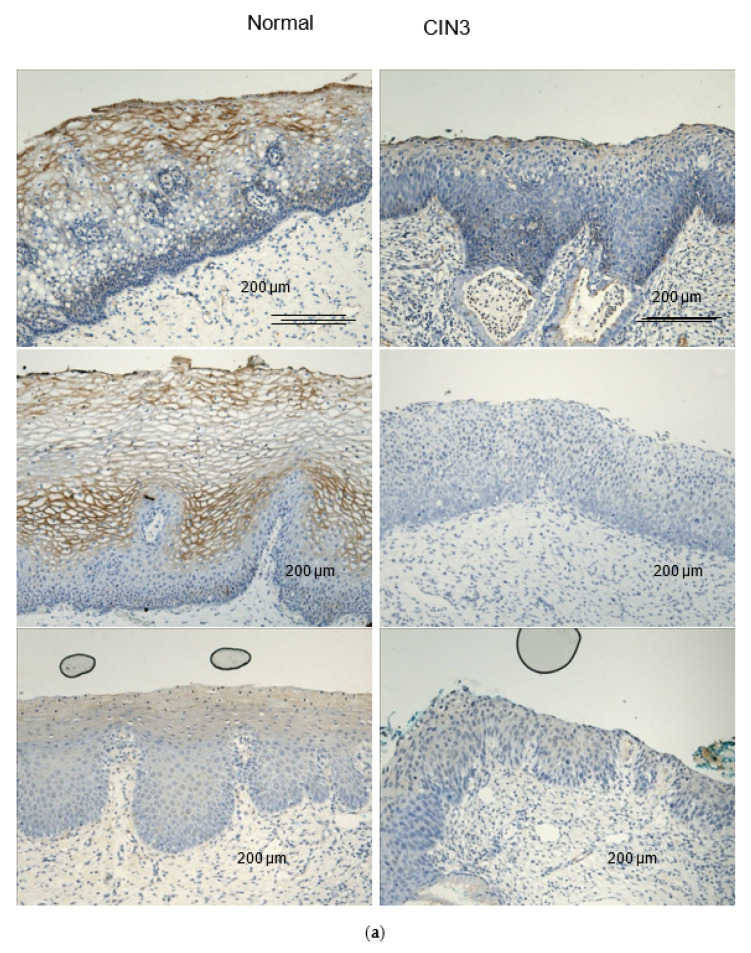
Lectin histochemistry for the discovery cohort for the three lectins which showed significantly weaker staining in CIN3 vs. normal human FFPE cervical tissue: WGA, HPA and UEA. (**a**) Representative examples of lectin histochemistry (brown). (**b**) Lectin staining scores (based on visual assessment) for (*n* = 20) normal versus (*n* = 20) CIN3 human FFPE cervical tissue samples for WGA, HPA and UEA.

**Figure 2 cancers-12-01928-f002:**
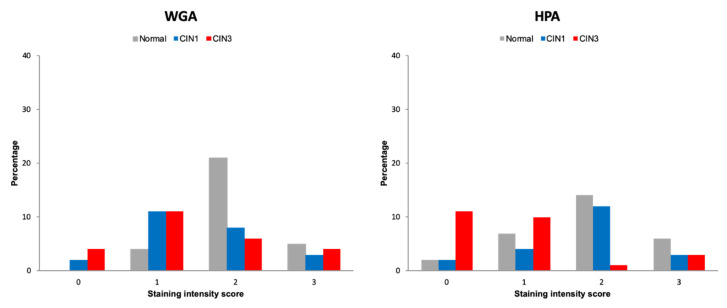
Frequency (%) of lectin histochemistry visual intensity staining scores for WGA and HPA for normal cervix, CIN1 and CIN3 human FFPE tissue in the validation cohort.

**Figure 3 cancers-12-01928-f003:**
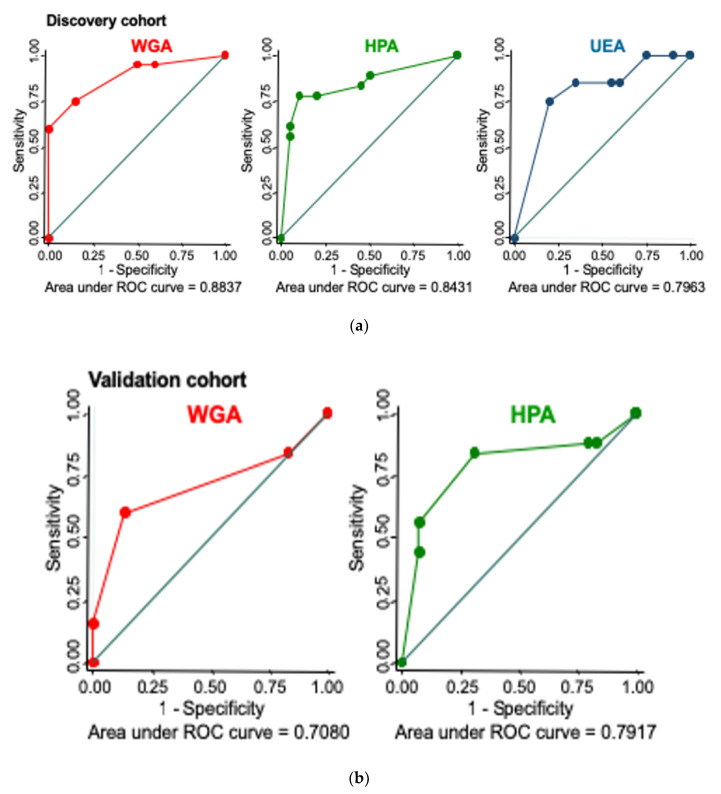
Receiver–Operator Curves for lectins with differential binding patterns in normal cervix versus CIN3 on lectin histochemistry using a threshold score of 0–1 = positive, and 2–3 = negative. (**a**) Discovery cohort (20 normal, 20 CIN3) showing ROCs for WGA, HPA and UEA. (**b**) Validation cohort (WGA: 30 normal, 25 CIN3, HPA WGA: 29 normal, 25 CIN3) ROC curves showing comparatively weaker discriminatory power for both WGA and HPA.

**Figure 4 cancers-12-01928-f004:**
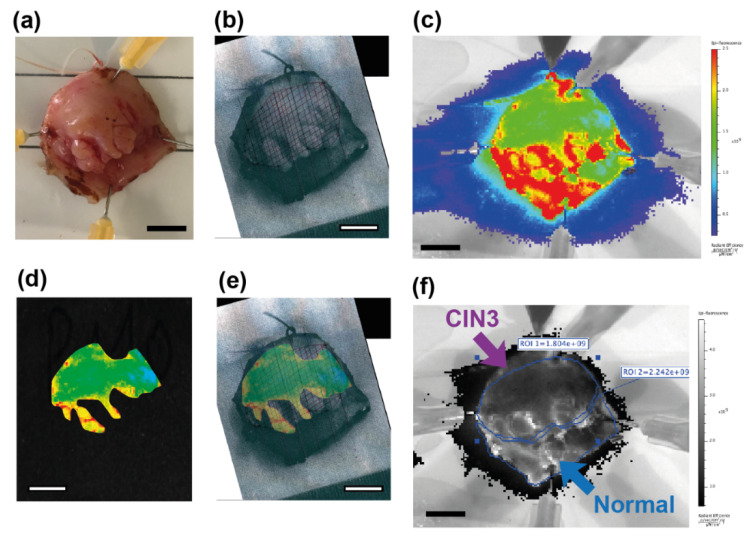
Images of a freshly resected human cervical specimen incubated with fluorescently-labelled WGA, which was selected for having a large area of CIN3 (**top**) surrounded by normal squamous epithelial tissue (**lower**). This specimen showed clear discrimination between CIN3 and normal squamous tissue. (**a**) White light image of the specimen; (**b**) specimen overlaid with histopathology grid with a crisscrossed pattern indicating a large area of CIN3 (top); (**c**) NIR fluorescence image displayed in pseudo-colour scale; (**d**) NIR fluorescence image of the same specimen which showed that the area of CIN3 has low fluorescence; (**e**) overlay image of the NIR fluorescent image with the corresponding pathology grid; (**f**) NIR fluorescence image (greyscale) showing that the lowest fluorescent signal intensity (ROI 1; top) is seen in the area of CIN3. Scale bar is 1 cm.

**Table 1 cancers-12-01928-t001:** Summary of freshly-resected whole-specimen human cervix samples for ex vivo molecular imaging using fluorescently-labelled WGA lectin.

Patient Number	Age (Years)	Pathological Diagnosis	Mean Fluorescent Intensity MFI (SD)		Contrast (%) ^1^	Dysplasia, % of Area
			Normal	CIN3		
101	53	Normal	0.111 (0.009)	N/E	N/D	0
104	38	CIN3	0.119 (0.008)	0.090 (0.008)	24.6	4.8
105	44	CIN3	0.063 (0.006)	0.0584 (0.004)	7.6	26.7
106	42	CIN3	0.184 (0.021)	0.146 (0.030)	20.8	18.6
107	28	CIN3	0.048 (0.006)	0.038 (0.007)	21.7	10.1
108	25	CIN3	0.153 (0.055)	0.115 (0.039)	24.8	49.3
110	25	CIN3	0.224 (0.073)	0.180 (0.055)	19.5	33.2

^1^ Defined as (MFI_CIN3_–MFI_normal_)/MFI_normal_ × 100 (%); N/D, not defined; N/E, non-existent.

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
