# Peer review of "Lectins in Cervical Screening"

_cancers, 2020, doi:10.3390/cancers12071928_

Round 1

Reviewer 1 Report

This paper was designed as a proof-of-principle study to investigate the availability of lectins as a new alternative to the existing inspection method for cervical cancer. It showed that, in topically applied, fluorescently-labelled lectins, it can be used to detect high-grade CIN as a form of cervical screening.

Lectins are naturally occurring glycoproteins, so many lectins have been applied in different areas of medical research and therapy because of their carbohydrate-binding specificities. In this trend, lectin-based approaches have been increased as a tool for cancer screening or novel cancer biomarkers. Many studies have been conducted in various other carcinomas, to investigate that lectin is a biomarker, but this study is the paper that presents a specific method for using lectin in screening for cervical cancer.

Although it is meaningful in that it suggested an alternative method for screening, I recommend that this paper needs to be revised to be accepted.

Major comment

  1. Lack of grounds for background explanation

 The rationale for considering lectins as a screening method in the background description is highly subjective of the existing visual inspection. However, the method of using lectin presented in the text is also entirely subjective (although it is mentioned in the limitation).

  1. Lack of grounds for Discussion and Conclusions

In common sense, the argument that 'lectin histochemistry' can be an advantage in low resource countries as a cost-effective screening method is not well understood. To be based on price rationality, it seems necessary to provide information on specific costs. There are many other tools in lectin-based techniques, and among them, an Enzyme-linked lectin assay (ELLA) is more cost-effective than the lectin histochemistry. If you want to highlight the advantages of price, it would be better to use ELLA as an analytical method rather than lectin histochemistry or to explain specific reasons using lectin histochemistry (Reference> Hashim et al. (2017), Lectins: an effective tool for screening of potential cancer biomarkers. PeerJ 5:e3784; DOI 10.7717/peerj.3784). Also, the accessibility of lectin histochemistry-related equipment can be a difficult problem to solve in low resource countries.

Author Response

  1. Lack of grounds for background explanation

Response 1

The rationale for considering lectins as a potential cervical screening tool is (1) that existing visual inspection methods are not very good (i.e. has variable sensitivity and low specificity) and therefore could be improved; and (2) there is evidence that lectin-binding changes with dysplasia/carcinogenesis in several organs, as written in the introduction:  

 “Neoplastic transformation of cells is associated with changes in cell surface glycoconjugates[7,8] and alterations in lectin binding with dysplasia or carcinogenesis have been observed in numerous organs”; “Lectins are naturally occurring glycoproteins of non-immune origin which bind to sugars (glycans) on the surface of cells.[6]

We have now amended the Introduction to make the rationale for investigating lectins as a cervical screening tool clearer by removing the word “subjectivity” and describing it as having room for improvement.

2. Lack of grounds for Discussion and Conclusions

Response 2

It is not our intention to develop lectins for use as a histological stain. The lectin histochemistry in this study was only performed as part of the discovery phase in which we were trying to determine whether or not lectins showed any discriminatory power in order to justify further exploration in ex-vivo studies. The use of lectins in cervical screening would be as a topically applied fluorescent tag or similar to allow real time visualisation for visual inspection.  Assays such as 'lectin histochemistry' or ELLA would be very hard to perform in loco and in real time in low resource countries. They would certainly require access to specialised equipment and laboratory facilities which are rarely available (e.g. spectrophotometer, plate reader, reagents, etc).

We propose to use lectins as a form of real-time visual assessment, using a fluorescence-capable light source (such as a hand held torch), coupled to an optical imaging device (such as a digital camera, colposcope or smartphone) to ‘illuminate’ areas of tissue with early disease and enable detection of molecular level changes. To further clarify this we have added the following new text to the Introduction (in blue)

The aim of this study was to act as a proof-of-principle study to explore the potential for using topically applied, fluorescently-labelled lectins to detect high-grade CIN as a form of cervical screening in low-resource settings. This approach would comprise real-time visual assessment using wide-field fluorescence imaging or fluorescence-enhanced colposcopy, based on a fluorescence-capable light source (e.g. torch).[27, 28]”  

Reviewer 2 Report

The authors present a study on using lectins (WGA) for cervical screening and to judge the presence or absence of CIN.

Although the number samples is very limited I see, that this work has to be sseen as a proof-of concept study. I think this is interesting work and worthwhile to publish after including some additional information.

Comments:

  • There should be more explanation on why the lectins may or may not be used for detection of CIN. Why is WGA working and others not. How is the impact and change of cell surface glycosylation due to CIN.
  • A statement should be added why lectins with similar specificty (GS-II/WGA) are not equally good for screening
  • The authors select certain lectins. It is unclear for me, which criteria the used, to select or unselect lectins. Again more explanation on specificty for certain glycans should be included.
  • The authors mention a contary study with GS-II Ref. 26,27. They should go more into detail on this discrepancy, as the other cited references in accord are not that similar to the present study.
  • Please improve the quality of the figures. Especially Figure 3 is blurred.
  • In section 4 the authors describe thus usage of a "lectin" buffer for the histochemical analyis including divalent cations important for lectin binding. Thus the ex vivo studies were accomplished with PBS without divalent cations. Is lectin binding comparable in both buffers?
  • It would be beneficial to estimate, if the interactions between lectins and samples is really glycan-driven. An competetive test using soluble carbohydrate could be performed.
  • I honestly doubt, that this system is really suitable for a "low-resource" seeting. Maybe the authors can either elaborate more, how fluorescence analysis can be "low-ressource" especially as a quite high sensitivity is needed or "soften" this aspect a little.

Author Response

Comments:

  • There should be more explanation on why the lectins may or may not be used for detection of CIN. Why is WGA working and others not. How is the impact and change of cell surface glycosylation due to CIN.

Response

The reviewer raises an important point here. The rationale for the use of lectins in general has been presented in the Introduction as:

Neoplastic transformation of cells is associated with changes in cell surface glycoconjugates[7,8] and alterations in lectin binding with dysplasia or carcinogenesis have been observed in numerous organs”; “Lectins are naturally occurring glycoproteins of non-immune origin which bind to sugars (glycans) on the surface of cells.[6]

However, no explicit reasoning was given for the use of WGA in the context of early cervical cancer detection. We have added the following to the Introduction.

“Aberrant sialylation and fucosylation have also been described as early markers of cervical carcinogenesis.[20]

“In cervical cancer, markedly reduced sialylation was found in the glycoproteins of malignant tissue.[20]

  • A statement should be added why lectins with similar specificty (GS-II/WGA) are not equally good for screening

Response

Indeed there are other lectins that show specificity for sialic acids (e.g. GS-II, TJA, SNA, MAL). However, their reduced availability and potential toxicity make them less appealing as candidates for topical imaging application than WGA, which is derived from edible wheat germ, a dietary constituent. We have added the following statement to section 2.3 of the Results

Other lectins have been used as histochemical reagents for sialic detection (e.g. GS-II, TJA, SNA, MAL). However, their reduced availability and potential toxicity make them less viable as candidates for topical imaging applications in comparison with WGA, which is derived from edible wheat germ, a dietary constituent.”

  • The authors select certain lectins. It is unclear for me, which criteria the used, to select or unselect lectins. Again more explanation on specificty for certain glycans should be included.

Response

The criteria for lectin selection has now been expanded upon in the Results section (see above and attached tracked revised manuscript).

  • The authors mention a contary study with GS-II Ref. 26,27. They should go more into detail on this discrepancy, as the other cited references in accord are not that similar to the present study.

Response

These studies, however, were performed with a majority of specimens containing CIN I-II, and very limited number of specimens containing CIN III. We have mentioned this in the Discussion now.

These studies, however, were performed with a majority of specimens containing CIN 1-2.”

  • Please improve the quality of the figures. Especially Figure 3 is blurred.

Response

We have now updated the figures and provided these as separate TIFF files for figures 1-3.

  • In section 4 the authors describe thus usage of a "lectin" buffer for the histochemical analysis including divalent cations important for lectin binding. Thus the ex vivo studies were accomplished with PBS without divalent cations. Is lectin binding comparable in both buffers?

Response

The reviewer is correct in mentioning the use of a “lectin buffer” for histochemical assays, in which divalent cations are present (e.g. Mg2+, Mn2+, Ca2+) and important for some of the lectin binding. However, in the case of WGA, the presence of such divalent ions is irrelevant as this lectin lacks any metal ion binding co-factors (see below) and thus binding is identical in the absence of such ions. Aiming at an in situ topical application in the context of a future clinical imaging study, a simpler physiological buffer formulation (e.g. PBS) is preferable.

“WGA is a homodimer composed of subunits of Mr = 23 600 Da which dissociates into monomers at acid pH. It is a pure, metal-free protein devoid of carbohydrate residues, which is isolated as a mixture of four isolectins …”.  

In https://www.sciencedirect.com/topics/biochemistry-genetics-and-molecular-biology/lectin

  • It would be beneficial to estimate, if the interactions between lectins and samples is really glycan-driven. An competetive test using soluble carbohydrate could be performed.

Response

We have performed such displacement tests in the context of oesophageal cancer specimens [ref. 9]. We haven’t been able to perform these displacement tests here, due to the reduced availability of specimens and time handling constraints imposed by ethics. Nevertheless, previous studies have shown that extract of cervical mucins are capable of inhibiting the blood agglutinating activity of WGA, suggesting high specificity of the lectin for sialic acid

[ref. https://www.jbc.org/content/254/10/4000.full.pdf ]

Similarly, WGA binding to endometrial adenocarcinoma has been shown to be displaced by its inhibitory saccharide - N-acetyl-glucosamine.

[ref. https://academic.oup.com/ajcp/article-abstract/82/3/259/1809269?redirectedFrom=PDF ]

  • I honestly doubt, that this system is really suitable for a "low-resource" seeting. Maybe the authors can either elaborate more, how fluorescence analysis can be "low-ressource" especially as a quite high sensitivity is needed or "soften" this aspect a little.

Response

We maintain that lectins have good potential to be used in low resource settings. Wheat Germ Agglutinin is widely available and cheap, and wide-field fluorescent imaging of lectins in this context would not involve a high cost laboratory test or expensive equipment. At the very least, there is no reason to believe that lectin-based optical imaging cervical screening would be more expensive than any of the currently available cervical screening tests. Furthermore, Baeten et al are currently developing a similar system for screening of oral neoplasia in India, using a simple hand-held fluorescent light, and one would hope that a similar system could be translated to cervix.

doi: 10.1016/j.joms.2019.03.012

doi: 10.1002/hed.24943

doi: 10.1016/j.tranon.2014.02.006

The latter point has been set out in the discussion:

“This model in oral cancer is highly relevant to the future development of lectins as a probe for cervical screening given that both malignancies arise in squamous epithelium and are particularly prevalent in developing countries in sub-Saharan Africa and South East Asia.[31, 32] Furthermore, a handheld light device is likely to be easily translated into cervical screening for low-resource settings where lightweight mobile equipment is desirable.[33]

Similarly, the emerging developments in cell-phone based wide field fluorescent microscopy also have potential to be used to translate lectins for cervical screening in low-resource settings. See DOI: 10.1109/IEMBS.2011.6091677

Given that both reviewers have expressed reservations about the cost-effectiveness and the potential for lectins to be used in low resource settings, we have cut-back the section in the discussion about cervical screening in low resource settings in order to soften this aspect.  We have also replaced the term “cost-effectiveness” with “affordable”.  

Round 2

Reviewer 1 Report

Information about HPV results and PAP smear of patients' are needed.

In validation set, more patients are needed to assess of sensitivity and specificity to high grade cervical disease.

As you mentioned, lectin histochemistry staining was calculated by pathologist, was there no discrepancy in that results? 

Author Response

1- Information about HPV results and PAP smear of patients' are needed.

In this study, H&E stained histology was the gold-standard that we were trying to approximate.  Including HPV and cytology results would not be informative in this context as the endpoint of CIN2+ is definitive and HPV status and cytology are merely correlates of CIN2+. The goal of cervical screening is to identify and treat high grade cervical lesions (CIN2+) so they can be removed to prevent cancer from developing.  Furthermore, we are proposing lectins for use in the screen and treat context where one is seeking to be able to distinguish CIN2+ lesions on the spot in order to remove them and almost all such lesions would be HPV positive and, (provided good quality samples were collected), cytology abnormal.

2- In validation set, more patients are needed to assess of sensitivity and specificity to high grade cervical disease.

This study was designed as proof of principle, i.e. to prove that there is a “signal” and that lectin staining of normal tissue and high-grade cervical dysplasia are not identical. We were not trying to show that the lectins approach is good enough to become a practical screening test or indeed to demonstrate that we have found a good screening test, but merely whether or not there is something that warrants further research.  Therefore, we maintain that the number of samples (patients) included at each stage of this proof-of-principle study are sufficient and appropriate. Furthermore, the validation set sample size was determined and justified a priori in the original study protocol.

We have now added additional text in the introduction and conclusion to emphasise the aims of the study were to show a “signal” rather than to investigate or define a new screening test (blue text below).

“The aim of this study was to act as a proof-of-principle study to explore whether fluorescently-labelled lectins, topically applied to the cervix show potential to be developed as a form of cervical screening in low-resource settings.”

“To our knowledge this is the first evidence that topically applied lectins show potential to be developed as a visual assessment tool for cervical screening. Our findings warrant further evaluation in an in vivo setting.”

3- As you mentioned, lectin histochemistry staining was calculated by pathologist, was there no discrepancy in that results?

There were discrepancies. A histopathologist adjudicated when this happened, and we generated a consensus result as described in section 4.4.1:

“For the discovery work, two researchers (including at least one histopathologist, N.S, S.L.L and P.L.S) independently scored the top third of squamous epithelium for intensity of lectin staining. A histopathologist (M.S) adjudicated when there was disagreement.”

We have now calculated the concordance between the two raters and the associated weighted Cohen’s kappa statistic and added this to the methods and results: 86.7% (kappa 0.65 [95%CI 0.60 – 0.67], p<0.0001)).

We have also provided a table below for Reviewer 1 for further detail. This shows the proportion of scores with a difference between rater 1 and 2 of 0, 1, 2 and 3 “units”.

Difference in score between raters

Percent

0

60.9%

1

35.5%

2

3.7%

3

0%

Round 3

Reviewer 1 Report

The manuscript was significantly improved.